# EMPIRICAL CONFIDENCE ESTIMATES FOR CLASSIFICATION BY DEEP NEURAL NETWORKS

## ABSTRACT

How well can we estimate the probability that the classification predicted by a deep neural network is correct (or in the Top 5)? It is well-known that the softmax values of the network are not estimates of the probabilities of class labels. However, there is a misconception that these values are not informative. We define the notion of *implied loss* and prove that if an uncertainty measure is an implied loss, then low uncertainty means high probability of correct (or top $k$) classification on the test set. We demonstrate empirically that these values can be used to measure the confidence that the classification is correct. Our method is simple to use on existing networks: we proposed confidence measures for Top $k$ which can be evaluated by binning values on the test set.

## 1 INTRODUCTION

Despite lots of effort to build confidence measures for classification by deep neural networks, there is still a lot of confusion about the value and applicability of these measures. In this article we present a simple method for estimating confidence based on *implied loss* values which leads to results which are empirically more accurate than benchmarks on test sets. We prove that high confidence values imply a high probability of correct classification on test sets.

Many have observed that used blindly, the maximum softmax probability of a network does a poor job of predicting uncertainty (Nguyen & O'Connor, 2015; Provost et al., 1998; Nguyen et al., 2015; Yu et al., 2011; Lakshminarayanan et al., 2017). However, Zaragoza & d'Alché Buc (1998) showed in the 1990s that on on shallow networks the maximum softmax probability and the (negative) entropy of the probabilities strongly correlate with model confidence on in-distribution images. More recently, in the deep setting, Hendrycks & Gimpel (2017) showed empirically that the maximum softmax probability can be used to predict network confidence. We demonstrate that both the maximum softmax and the model entropy are uncertainly measures. We extend the uncertainty metric to Top $k$ predictions. We show that, in conjunction with binning, simple uncertainty statistics outperform common approaches like MC-dropout as a measure of confidence, at a fraction of the computational cost.

Using this simple idea, we make the following contributions.

1. We estimate the probability that the classification of the model on a test set is correct which works for existing models (no need to retrain), using a simple tabular form (see Table 2 for Imagenet).

2. We give a simple definition of uncertainty, which applies to previously proposed methods, and leads to a proof that low uncertainty (high confidence) implies high probability of correct classification. It applies to both Top 1 and Top $k$ uncertainty.

3. We can discover mislabelled data, see Figure 1(c) and we can detect off manifold data and adversarial examples.

We advocate evaluating model uncertainty via expected *Bayes factors* (Kass & Raftery, 1995), which provide a rigorous probabilistic approach to evaluating uncertainty, and are widely used for hypothesis testing in other scientific fields, see for example (Good, 1979) and (Jeffreys, 2003). Compared to other methods (such as AUROC or Brier scores) Bayes factors better distinguish improvements to

confidence for methods which are already quite accurate, as is the case for top 1 or top 5 uncertainty for image classification.

## 2    PRIOR WORK

As neural networks are adopted into safety critical systems, the need for neural network uncertainty estimates has become abundantly clear. Indeed, any accident adverse system must by design incorporate notions of uncertainty (Amodei et al., 2016). Real-world examples abound: uncertainty measures are needed in autonomous vehicles (Feng et al., 2018), robotics (Richter & Roy, 2017), medical imaging (Ching et al., 2018; DeVries & Taylor, 2018b) and medical decision making (Begoli et al., 2019), and semantic understanding (Kendall et al., 2017).

Much effort has been dedicated to addressing this deficiency. Many works have placed neural networks within a Bayesian probabilistic framework. Initial work placed Bayesian priors on model weights (MacKay, 1992a; Neal, 1996), leading to Bayesian neural networks, however this has proven difficult to implement in practice. Many techniques have been developed to overcome this difficulty (MacKay, 1992b; Neal, 1996; Graves, 2011; Hasenclever et al., 2017; Li et al., 2015; Balan et al., 2015; Welling & Teh, 2011; Springenberg et al., 2016). One promising approach in the deep learning setting is to perform *approximate* posterior inference (Louizos & Welling, 2016; Hernández-Lobato & Adams, 2015; Blundell et al., 2015; Sun et al., 2017).

Due to its simplicity, dropout is widely used as a surrogate for uncertainty. Dropout (Srivastava et al., 2014) was interpreted in a Bayesian setting by Gal & Ghahramani (2016) and Kingma et al. (2015), however, there are problems with this interpretation, see (Hron et al., 2018) for a recent discussion. Dropout involves evaluating an ensemble of models at test time, which can be both memory and computationally intensive for very large networks.

Non-Bayesian model ensembles have also been developed (Dietterich, 2000), for a recent survey see (Li et al., 2018). Lakshminarayanan et al. (2017) train an ensemble of adversarially robust models and empirically showed an improvement in uncertainty estimates over dropout based methods. Geifman et al. (2018) proposed using an early stopping criteria to collate an ensemble of models. Kristiadi & Fischer (2019) use mixture modeling to chose ensemble weights.

Several deep learning specific approaches have been proposed in recent years, especially in the context of detecting out-of-distribution samples. Oliveira et al. (2016) suggest detecting outliers via an anomaly detector. Lee et al. (2018) generator out-of-distribution images through a GAN; the classifier is trained to assign the equal weight probability vector to these images. Hendrycks et al. (2018) train networks on two distributions: the in-distribution samples, and out-of-distribution samples. Liu et al. (2018) develop PAC-style guarantees on detection of out-of-distribution samples. Several recent works (Jiang et al., 2018; Papernot & McDaniel, 2018; Mandelbaum & Weinshall, 2017) have suggested using nearest neighbour distances, in feature space, for outlier detection and confidence measures. DeVries & Taylor (2018a) suggest training an additional network to predict uncertainty; Malinin & Gales (2018) specifically model prediction probabilities with a Dirichlet distribution, which implicitly describes model uncertainty.

Platt (2000) proposed scaling SVM predictions to better match the validation set; this has been generalized to neural networks and multiclass classification (Niculescu-Mizil & Caruana, 2005; Guo et al., 2017). Other scaling approaches, such as changing the softmax temperature, have shown promise (Guo et al., 2017; Liang et al., 2018). Another popular approach to calibration is based on *binning* model probabilities, developed by Zadrozny & Elkan (2001). Each bin is assigned a probability of being correct, which is obtained by minimizing the Brier score of the bins (Brier, 1950). Bins edges may be optimized as well (Zadrozny & Elkan, 2002); and can be extended to the Bayesian setting by assigning a prior on binning schemes (Naeini et al., 2015).

## 3    CONFIDENCE MEASURES AND UNCERTAINTY ESTIMATES

Suppose a model $f(x)$, generalizes well, so that it has a high probability, $p$, of a correct prediction on an image $x$ sampled from the same underlying distribution. Write $I_k(f) = \{$indices of the $k$ largest components of $f\}$ for the top $k$ indices. The classification of the vector $f$ is given by the largest component, $C(f) = I_1(f)$. Define the random variables $X_k = 1_{\{y(x) \in I_k(f(x))\}}$

to be the Bernoulli random variables with expected value $p_k = \mathbb{E}[X_k]$, the probability that the correct label is in the Top $k$. Our goal is to estimate $p_k$. We do this by defining random variables, $U_k$, which we call *uncertainties*, whose statistics allow us to better estimate $p_k$. The histogram of the uncertainty variables will result in an estimate of the conditional probability that the classification is correct, given the uncertainty value,

$$\text{Prob}\left(X_k(x) = 1 \mid U_k(x) = t\right).$$

**Definition 3.1.** Given $\epsilon > 0$, and the uncertainty measure $U(x)$, define the set

$$S_k^\epsilon = \{U_k(x) \leq \epsilon \text{ and } y \notin I_k(x)\} \tag{1}$$

The uncertainty measure $U_k(x)$ is an implied loss if the event $S_k^\epsilon$ has high expected loss.

*Example* 3.2. For the Kullback-Leibler loss, the (negative) entropy of the probabilities is an uncertainty measure. Write $f^{sort}$ for the indices of $f$ sorted in decreasing order. Define

$$U_1(x) = -\log(f_1^{sort}), \qquad U_k(x) = -\log\left(\sum_{i=1}^{k} f_i^{sort}\right), \tag{2}$$

*Example* 3.3. For general losses, $\mathcal{L}$, the top 1 uncertainty can be given by $U_1(x) = \{\mathcal{L}(f(x), y) \mid y = C(f(x))\}$, which is the *loss, given that the classification was correct*. Also $U_k(x) = \mathcal{L}(f, y_w)$ is an implied loss, for top $k$, where $y_w$ is the $(k+1)$-th ranked label.

### 3.1 ILLUSTRATION OF UNCERTAINTY RANDOM VARIABLES

The histogram of $U_1$ is plotted on the test set in Figure 1(a). Note that for small values of $U_1$, the images have a very high probability of being correct. In fact, we can use $U_1$ to detect incorrectly classified images: we visualized the images which smallest value of $U_1$ (i.e. highest confidence), which correspond to the few isolated points in the upper left of the figure. It turned out that all of these were either incorrectly labelled, or were ambiguous images, see illustrations in Figure 1(c). For example, in the second image, the animal is a wallaby, not a wombat. In the fourth image, a paintbrush is a kind of plant, but there is also a pot in the image. In the second part of Figure 1(b) we illustrate more quantitatively the Top 1 (green) and Top 5 (green or blue) probabilities conditioned on the 100 histograms bins of $-\log(p_{\max})$ on test set for ResNet152 on ImageNet. The Top 1 probability conditioned on the lower bins is very close to 100%. The Top 5 probability is no better than 50% on the last few bins. The intermediate bins are less informative.

### 3.2 TOP 1 UNCERTAINTY ESTIMATES

The next theorem shows that if the uncertainty is small, then the probability of correct classification must be high.

**Theorem 3.4** (Confidence estimate)**.** *Define $U_1(x)$ by (2) and define $S^\epsilon$ by (1), and let $\mathcal{L}_{KL}$ be the Kullback-Leibler loss. Then*

$$\text{Prob}\left(S^\epsilon\right) \leq \frac{\mathbb{E}\left[\mathcal{L}_{KL}(f(x), y)\right]}{\log\left(\frac{1}{\epsilon}\right)} \tag{3}$$

*Proof.* Claim: Let $\epsilon > 0$ be small. By assumption, $-\log f_1^{sort} \leq \epsilon$. Thus $f_1^{sort} \geq \exp(-\epsilon)$. Let $e_k$ be the correct label. Then $f_k \leq f_1^{sort}$, so

$$f_k \leq 1 - \exp(-\epsilon)$$

and

$$-\log(f_k) \geq -\log(1 - \exp(-\epsilon)) \geq \log(1/\epsilon).$$

Thus for $x \in S^\epsilon$, $\mathcal{L}_{KL}(f(x), y(x)) \geq \log(1/\epsilon)$. Apply Markov's inequality (13) to the random variable $L(x) = \mathcal{L}(f(x), y(x))$ to obtain the result. $\square$

*Remark* 3.5 (Neural Networks are always overconfident)**.** Note that the uncertainly is always less than the loss,

$$U_1(f) \leq \mathcal{L}_{KL}(f, e_k) \tag{4}$$

with equality when $C(f(x)) = y(x)$.

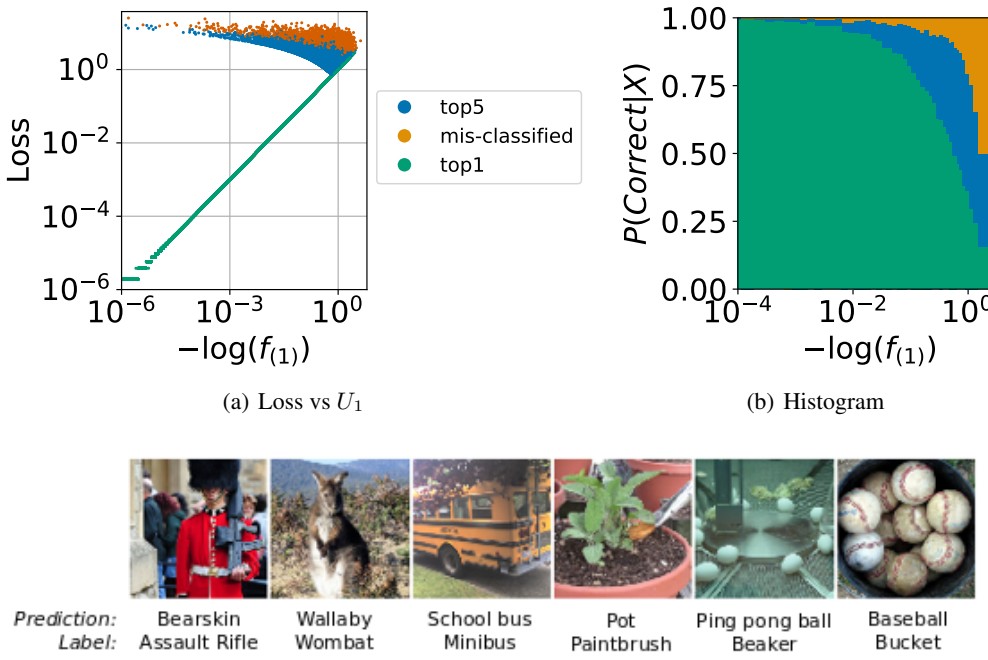

(a) Loss vs $U_1$

(b) Histogram

(c) mislabelled or ambiguous images found using $U_1$.

Figure 1: Figure 1(a) Scatter plot to indicate how predictive $U_1$ is compared to the loss. For small values of $U_1$, the loss is small with high probability. Figure 1(b): the probability correct (green) or Top5 (blue) given the value of $U_1$. Figure 1(c): visualization of the images in the upper left of Figure 1(a). The confident images which were labelled incorrectly turned out to be mislabelled or ambiguous.

### 3.3 TOP $k$ UNCERTAINTY ESTIMATE

In the next result we show that if the top $k$ uncertainty is small, then the probability that the correct labels is in the top $k$ must be high. The result can also be proven in the case of general losses, and uncertainly measures satisfying (1).

Consider the event $S_k^\epsilon$ (1) for a given $k \geq 1$. If the correct label is not in the top $k$, then the probability of the correct label, $f_c$, must satisfy

$$f_c \leq f_{k+1}^{sort}$$

with

$$f_{k+1}^{sort} \leq 1 - (f_1^{sort} + \cdots + f_k^{sort})$$

Thus

$$\mathcal{L}_{KL}(f, e_c) \geq -\log(1 - (f_1^{sort} + \cdots + f_k^{sort}))$$

Then, by an argument similar to the one for Top 1 error, we see that

$$\text{Prob}\left(S_k^\epsilon\right) \leq \frac{\mathbb{E}\left[X_k\right]}{\log\left(\frac{1}{\epsilon}\right)} \tag{5}$$

## 4 EMPIRICAL RESULTS

The previous section proved that, under fairly general conditions, we can define uncertainty measure which ensure that the top $k$ classification is correct with high probability. The theory applies to uncertainties used in the literature, such as the negative entropy of the probabilities, and negative log softmax.

In practice, once we have an uncertainty measure, the method is simple

1. Compute the statistics on the test set of the uncertainty estimates.
2. Divide the test set into bins, based on uncertainty values.
3. Estimate the conditional probabilities based on the bin populations.

## 4.1 THE BAYES FACTOR

The Bayes factor is a way to measure the value of new information, in terms of how much the expected winnings of a fair bet increase, when the information is available. Unlike other measures of confidence, which are additive, the Bayes factor is *multiplicative*. On a model which is correct 95% of the time, there still a lot of value in knowing when the probability correct increases to 99.5%. In this case, the Bayes factor is close to 10. On the other hand, going from 50% to 54.5% gives a Bayes factor close to 1.2. On the other hand, additive scoring methods give equal weight to both improvements. As an example, we show in Table 5 that the Brier scores of eight different measures of confidence all lie close together, between .033 and .076. On the other hand, the expected Bayes factors range more widely, from 1.3 to 16.6.

Consider a Bernoulli random variable $X = B(p_X)$. The odds for $X$ are given by $O(p_X) = \frac{p_X}{1-p_X}$. Now consider a test, $Y = B(p_Y)$, for which

$$p_{X,Y} = \text{Prob}\,(X = 1 \mid Y = 1)$$

Then the odds, given the test succeeds, are $O(p_{X,Y}) = \frac{p_{X,Y}}{1-p_{X,Y}}$. If the odds have increased, we define the Bayes Factor to be $BF(X \mid Y) = \frac{O(p_{X,Y})}{O(p_X)}$. On the other hand, if the odds have decreased, then the value of the information provided by $Y$ is to bet against, so we define the Bayes factor to be $BF(X \mid Y) = \frac{O(p_X)}{O(p_{X,Y})}$. Combining these possibilities, define the Bayes factor by

$$BF(X \mid Y) = \max\left(\frac{O(p_{X,Y})}{O(p_X)}, \frac{O(p_X)}{O(p_{X,Y})}\right) \tag{6}$$

## 4.2 EXPECTED BAYES FACTOR

We can also define a metric for measuring the quality of an uncertainty random variable, such as the value of the loss (or another random variable) for predicting the probability of correct classification. In practice, we will define the tests based on bin values for some uncertainty variable. If we have more data, we can use more bins.

**Definition 4.1** (Histogram random variables). Given a random variable $U(x) \in [a, b]$ and a partition of $[a, b]$ into bins

$$a = t_0 < t_1 \cdots < t_Q = b, \tag{7}$$

Define the histogram (or bin) random variables $B_i$, corresponding to each interval

$$B_i(x) = \begin{cases} 1 & t_{i-1} \leq U(x) < t_i \\ 0 & \text{otherwise} \end{cases} \tag{8}$$

so that

$$\text{Prob}\,(t_{i-1} \leq U < t_i) = \mathbb{E}\,[B_i] \tag{9}$$

Each Bayes factor measures the value of information that $x$ lies in each quantile. The value of the test itself is defined to be the expected value of the Bayes factors.

**Definition 4.2** (Histogram Bayes Factors). Given $X$, $U$ and the histogram random variables $B_i$, define the conditional probabilities

$$p_{X,i} = \text{Prob}\,(X = 1 \mid B_i = 1), \quad i = 1, \ldots, Q \tag{10}$$

The predictive value for $X$ of the random variable $U$ with respect to the histogram, is given by

$$\mathbb{E}\,[BF(X \mid Y_i)] = \sum_{i=1}^{Q} BF(X \mid B_i)\,\mathbb{E}\,[B_i] \tag{11}$$

When the number of bins is large, we defined the bins to be quantiles, so that they have an equal number of examples in each bin. In addition, in order to make the histogram Bayes factors finite, we require that each bin have at least one correct and one incorrect example.

See Appendix C for a detailed example of the expected Bayes factor. Next we present an example based on actual confidence measures for networks.

Table 1: Bayes factor $\mathbb{E}[BR]$ against various measures of confidence. For CIFAR-10 we used $X_1$, the probability of the correct label; for CIFAR-100 and ImageNet-1K we used $X_5$ the probability that the correct label is in the Top5. Data is binned into 100 bins, chosen to have equal weight.

| Confidence measure | CIFAR-10 | CIFAR-100 | ImageNet-1K |
|---|---|---|---|
| Model Entropy | 4.29 | 3.64 | 8.18 |
| $-\log p_{\max}$ | 4.22 | 3.77 | **8.87** |
| $-\log \sum p_{1:5}$ | - | **4.25** | 8.45 |
| $\|\nabla_x \|p\|\|$ | 8.32 | 3.47 | 7.17 |
| Dropout variance ($p = 0.002$) | 10.39 | 3.11 | 6.84 |
| Dropout variance ($p = 0.01$) | 4.67 | 2.38 | 7.81 |
| Dropout variance ($p = 0.05$) | 1.69 | 1.35 | 1.60 |
| Ensemble variance | **16.66** | 4.03 | 6.13 |
| Loss | $\infty$ | 228.94 | 1242.55 |

*Example* 4.3. This example follows closely the confidence bins for top 5 on ImageNet-1K, using the Model Entropy, as in the first row of Table 2. Consider a model with $p_X = .94$. In this the odds are 94 to 6, so $O(p_X) = 15.6$. Define three bins for Model Entropy with bin edges $0.31, .140$. Then with probability .55, data is in the first bin, in which case the probability correct is .99. So the Bayes factor in this bin is $(.99/.94)(.06/.01) = 6.3$. Thus knowing the Model Entropy is less than .31 tell you that you are 6.3 times more likely to be correct than on average. The second bin consists of data with with Model Entropy between .31 and .14, which occurs with probability .31, in this case, the Bayes Factor $(95/94)(6/5) = 1.2$ is nearly one, so there is little additional value to knowing data is in this bin. Finally, when the Model Entropy is greater than .14, which occurs with probability .14, the probability correct is only .8. In this case, the relative probability to correct is worse, to the Bayes Factor is given by $(94/80)/(6/20) = 3.9$. The expected Bayes factor is the weighted average of the Bayes factor of each bin, weighted by the probability of the bins

$$\mathbb{E}\left[BF(X \mid B_i)\right] = 6.3 \times .55 + 1.2 \times .31 + 3.9 \times .14 = 4.4$$

So the expected value of the Model Entropy, for the chosen bins, is 4.4. By fine graining the bins we can capture relatively small and relatively large values of the Model Entropy which can have Bayes factors on the order of 20, see Figure 5. Thus the expected Bayes factor with 100 bins is 8.18, as shown in Table 1.

### 4.3    Confidence bins and Bayes Factor

In this section we present confidence bins for ImageNet-1K. These bins are concise summaries of the information presented in the larger bins. Table 2 presents short bins for ImageNet. Using these bins, we can simply read of from the Uncertainty values, the probability that the model is correct. For example, on the model, $P(\text{top } 5) = 0.9406$, however, using entropy, 55% of the images had entropy low enough to be confidently classified with probability .99. Using $U_5$, 66% of images could be binned to have probability .99.

Bins for CIFAR-10 and CIFAR-100 are given in Tables 7 and 6, respectively.

In Table 1 we show the expected Bayes factor for various confidence measures, on CIFAR-10, CIFAR-100, and ImageNet-1K. In addition to the confidence measures already discussed, we considered Bayesian dropout, and the norm of the gradient of the model. Larger expected Bayes factors means the information is more valuable.

In Figure 5 we plot the regularized Bayes factor for our two main measure of confidence, $U_1$ and $U_5$ along with the loss and the model Entropy. The entropy and $U_1$, $U_5$ have very large Bayes factor in the first 10 and last 3 bins, meaning that for these bins, the prediction is 10X (or more) likely to be correct (for the first 10) or wrong (for the last 3 bins) than average.

Table 2: Confidence bins for ImageNet-1K. The values of $a$ and $b$ are chosen such that $P(\text{top5} \mid Y < a) = 0.99$ and $P(a \leq \text{top5} \mid Y < b) = 0.95$. For the model used here, $P(\text{top5}) = 0.9406$.

| Confidence measure $Y$ | $(a, b)$ | $P(Y < a)$ | $P(a \leq Y < b)$ | $P(Y \geq b)$ |
|---|---|---|---|---|
| Model Entropy | (0.31, 1.40) | 0.55 | 0.31 | 0.14 |
| $-\log p_{\max}$ | (0.047, 0.41) | 0.52 | 0.26 | 0.22 |
| $-\log \sum p_{1:5}$ | (6.2e$-$3, 0.03) | 0.66 | 0.13 | 0.21 |
| $\|\nabla_x \|p\|\|$ | (0.19, 0.30) | 0.52 | 0.08 | 0.40 |
| Dropout variance ($p = 0.002$) | (8.5e$-$4, 4.7e$-$3) | 0.50 | 0.15 | 0.35 |
| Ensemble variance | (0.014, 0.023) | 0.54 | 0.05 | 0.41 |

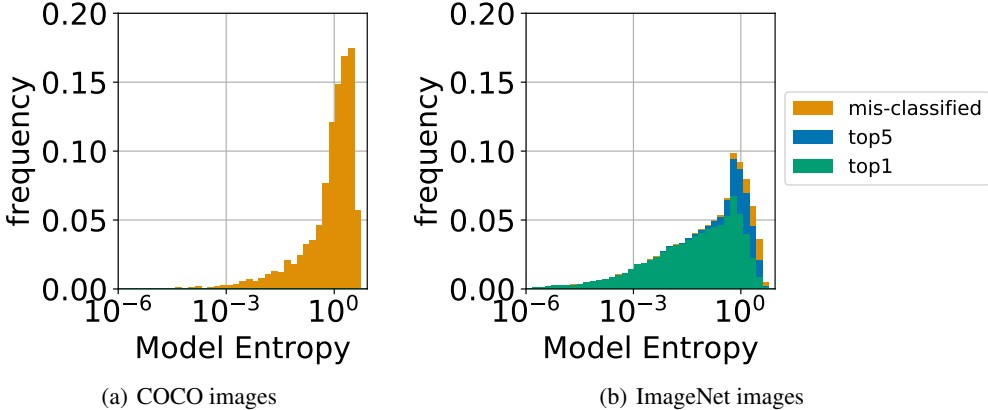

(a) COCO images          (b) ImageNet images

Figure 2: Figure 2(a): Confidence of a model trained on ImageNet-1k, evaluated on the COCO dataset. Figure 2(b): ImageNet images.

## 5 EXTENSIONS

In this section we discuss some extensions of the confidence results. We show that we can detect mislabeled images in the test set. We also show that we can obtain some confidence results for off manifold images, as well as adversarial images.

### 5.1 DETECTION OF MISLABELED IMAGES

We are able to detect test images which are mis-labeled: images which the network correctly classified, but who's label is incorrect, or for which multiple labels could apply. These are images with high loss but low model entropy. For example in Figure 1(c) we show six images from the ImageNet-1k test set who's predictions where not in the top5, but had low model entropy. All six of these images either have an incorrect dataset label, or could be described by multiple labels. For example, in the second image, the animal is a wallaby, not a wombat. In the fourth image, a paintbrush is a kind of plant, but there is also a pot in the image.

### 5.2 CONFIDENCE ON OUT-OF-DISTRIBUTION AND ADVERSARIAL IMAGES

Next we studied whether we could detect out-of-distribution images generated by COCO. In Figure 2 we show how the histogram of the model entropy is shifted to the right compared to the on-distribution images. Table 3 give the results of our test: choosing a confidence measure which rejects 10% of the on-distribution images, our confidence measures rejected as much as 38% of COCO images (for Entropy) with similar values for $U_1, U_5$. On the other hand Dropout was completely ineffective.

Table 3: Discarding out-of-distribution images from ImageNet-1K. For each confidence measure $Y$, the value of $a$ is chosen such that $P(Y \leq a \mid \text{image is from ImageNet-1k}) = 0.9$.

| Image source | Confidence measure | $a$ | $P(\text{image discarded})$ |
|---|---|---|---|
| COCO | Model Entropy | 1.75 | **0.38** |
| | $-\log p_{\max}$ | 0.77 | 0.34 |
| | $-\log \sum p_{1:5}$ | 0.13 | 0.37 |
| | $\|\nabla_x \|p\|\|$ | 1.06 | 0.23 |
| | Dropout variance ($p = 0.002$) | 0.024 | 0. |
| adversarially perturbed ($L_2$) | Model Entropy | 1.75 | 0.28 |
| | $-\log p_{\max}$ | 0.77 | 0.25 |
| | $-\log \sum p_{1:5}$ | 0.13 | 0.28 |
| | $\|\nabla_x \|p\|\|$ | 1.06 | **0.58** |
| | Dropout variance ($p = 0.002$) | 0.024 | 0.39 |

Table 4: Adversarial detection with ResNeXt-34 (2x32) on CIFAR-10. Clean images which the model correctly labels are perturbed until they are misclassified with four attack methods (PGD, Boundary attack, Carlini-Wagner, and an evasive Carlini-Wagner designed to avoid detection). Images are rejected if $|\nabla f(x)|_{2,\infty} > 2.45$.

| | clean | PGD | Boundary | CW | evasive CW |
|---|---|---|---|---|---|
| percent detected | 6% | 96% | 100% | 100% | 22% |
| median $\ell_2$ | - | 0.31 | 0.36 | 0.34 | 0.81 |

## 6 CONCLUSIONS

With the goal of using measures such as model entropy as a surrogate for the (unknown) model loss, we defined confidence measures as random variables which are large when the loss is large. Using this definition we proved that confidence variables can be used to estimate the probability that a model low expected loss makes a correct prediction. In practical terms, this amounts to defining a confidence measure (such as model entropy or log pmax) and binning the values.

We presented the expected Bayes factor as an effective measure of confidence. Since models are already very accurate, it is important to measure the relative confidence. The Bayes factor is a multiplicative factor to the probability (or odds) that a model is correct. For example, showing the that model entropy on ImageNet with 100 bins is 8 means that knowledge of the model entropy (and the bayes factors for the bins) allows us to predict the probability that the model is correct 8 times more effectively.

The Bayes factors was used to compare existing confidence measures on different tasks. The main task was estimating the probability of a correct prediction on the images from the data set. Additional tasks included: detection of off manifold data, detection of adversarial examples, and detection of mislabelled images. The latter were found by searching for highly confident predictions which were labelled incorrect.

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

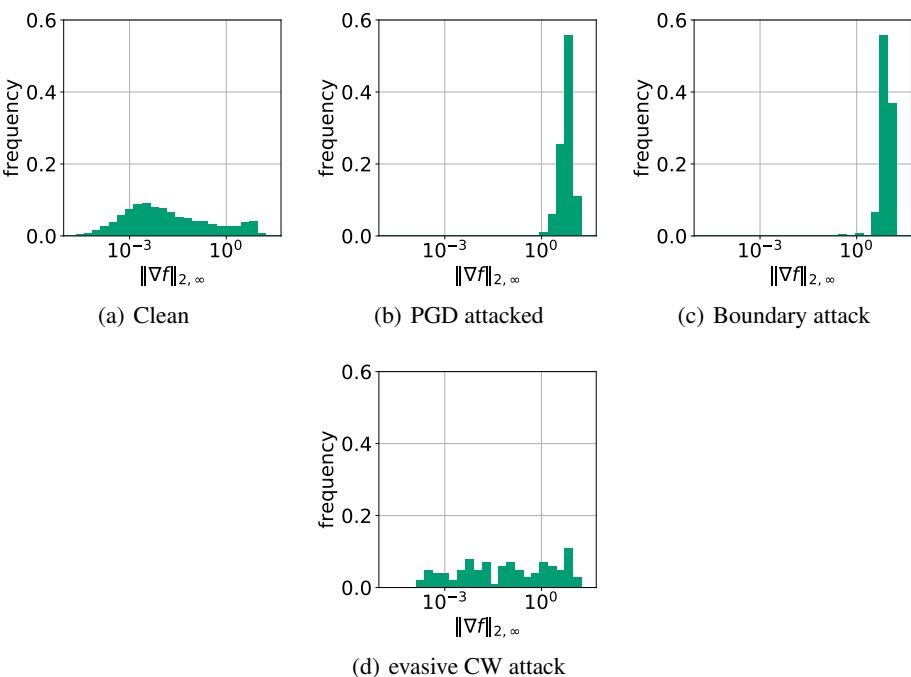

Figure 3: Frequency distribution of the norm of the model Jacobian $\left|\nabla f(x)\right|_{2,\infty}$ on ResNeXt-34 (2x32) on CIFAR-10, using 3(a): Clean, 3(b): PGD attacked 3(c): Boundary attacked, 3(d): evasive-CW attacked test images.

## A ADVERSARIAL ATTACK DETECTION

In this section we empirically demonstrate that image vulnerability may also be used to *detect* adversarial examples. We hypothesize that unless otherwise penalized, gradient based attacks will tend to move images to regions where the gradient of the loss is large. Based on this heuristic, we propose the norm of the loss gradient norm as criterion for detecting adversarial perturbations. Because the loss is not available during inference, we propose using the norm of the model gradient as a rejection criteria: an image has been adversarially perturbed if

$$\|\nabla f(x)\|_{2,\infty} \geq c, \tag{12}$$

for some threshold value, $c$. The threshold is determined by setting the significance level (the rate of false positives) to 5%. For example on CIFAR-10 we obtained $c = 2.45$ for our model. The results are reported in Table 4 and in Figure 3. Only 6% of clean test images were rejected. However, 100% of Boundary attacks and Carlini-Wagner attacks were detected, as well as 96% of PGD attacked images.

This leads to the question, is it possible to successfully perturb all images in the test set, and avoid detection? We built a targeted attack, designed to avoid detection. We use a Carlini-Wagner style attack, modified with a penalty to avoid detection. We augmented the attack loss function with a penalty for $\|\nabla \ell(x)\|_*^2$, which penalizes attacks for being detectable. We call this attack an evasive Carlini-Wagner attack. The evasive CW attack was successful at avoiding detection 78% of the time, but in order to do so, it increased the median adversarial distance significantly, from 0.31 to 0.81, see Table 4.

## B MARKOV'S INEQUALITY

**Lemma B.1** (Markov's Inequality). *For a random variable $Z$ with finite expectation, let $S \subset \{Z \geq a\}$ then*

$$\text{Prob}\left(S\right) \leq \frac{\mathbb{E}\left[Z\right]}{a} \tag{13}$$

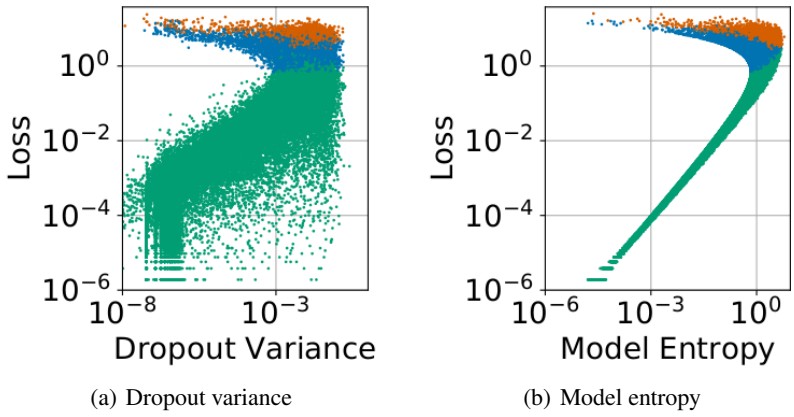

(a) Dropout variance

(b) Model entropy

Figure 4: Illustration of uncertainty measures on ImageNet. Dropout $p = 0.002$.

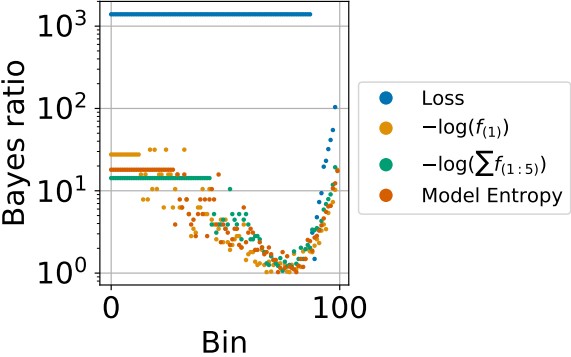

Figure 5: Bayes factor over equal 100 quantile bins on test set for ImageNet: loss, entropy, $U_1, U_5$. The entropy and $U_1, U_5$ have very large Bayes factors in the first 10 and last 3 bins.

Table 5: Brier score of various measures of confidence. For CIFAR-10 we used $X_1$, the probability of the correct label; for CIFAR-100 and ImageNet-1K we used $X_5$ the probability that the correct label is in the $\text{Top5}$. Data is binned into 100 bins, chosen to have equal weight.

| Confidence measure | CIFAR-10 | CIFAR-100 | ImageNet-1K |
|---|---|---|---|
| Model Entropy | **0.033** | 0.067 | 0.041 |
| $-\log p_{\max}$ | **0.033** | 0.067 | 0.042 |
| $-\log \sum p_{1:5}$ | - | 0.067 | **0.040** |
| $\|\nabla_x \|p\|\|$ | 0.034 | 0.073 | 0.046 |
| Dropout variance ($p = 0.002$) | 0.036 | 0.074 | 0.047 |
| Dropout variance ($p = 0.01$) | 0.04 | 0.075 | 0.048 |
| Dropout variance ($p = 0.05$) | 0.043 | 0.076 | 0.049 |
| Ensemble variance | 0.040 | **0.050** | 0.047 |
| Loss | 0 | 0.029 | 0.019 |

Table 6: Confidence bins for CIFAR-100. The values of $a$ and $b$ are chosen such that $P(\text{top5} \mid Y < a) = 0.99$ and $P(a \leq \text{top5} \mid Y < b) = 0.95$. For the model used here, $P(\text{top5}) = 0.916$.

| Confidence measure $Y$ | $(a, b)$ | $P(Y < a)$ | $P(a \leq Y < b)$ | $P(Y \geq b)$ |
|---|---|---|---|---|
| Model Entropy | (0.082, 2.1) | 0.24 | 0.50 | 0.26 |
| $-\log p_{\max}$ | (7.9e−3, 0.42) | 0.24 | 0.49 | 0.27 |
| $-\log \sum p_{1:5}$ | (4.8e−3, 0.34) | 0.19 | 0.57 | 0.24 |
| $\|\nabla_x \|p\|\|$ | (0.46, 1.70) | 0.27 | 0.17 | 0.56 |
| Dropout variance ($p = 0.002$) | (6.4e−4, 2.2e−3) | 0.27 | 0.06 | 0.67 |
| Ensemble variance | (4.2e−4, 0.052) | 0.42 | 0.18 | 0.40 |

Table 7: Confidence bins for CIFAR-10. The value of $a$ is chosen such that $P(\text{top1} \mid Y < a) = 0.975$.

| Confidence measure $Y$ | $a$ | $P(Y < a)$ | $P(Y \geq a)$ |
|---|---|---|---|
| Model Entropy | 1.6 | 0.95 | 0.05 |
| $-\log p_{\max}$ | 0.57 | 0.95 | 0.05 |
| $\|\nabla_x \|p\|\|$ | 8.16 | 0.93 | 0.07 |
| Dropout variance ($p = 0.002$) | 0.045 | 0.92 | 0.08 |
| Ensemble variance | 0.019 | 0.88 | 0.12 |

## C  WORKED EXAMPLE OF BAYES FACTORS

Consider the situation where you have exchanged phone numbers with someone, and you wish to contact them. The question is whether to send a text message or phone their number. Approximately 95% of people prefer to message. Let $X$ be the probability that a person prefers to message. The expected value and odds for $X$ is given by

$$p_X = 0.95, \qquad O(p_X) = 19$$

Now suppose we have additional information, which gives these statistics based on age. Suppose we wish to predict $X$. Knowing the age $U$ has a value. Let $U(x)$ be the age, and consider three bins for $U$ given by the values $20, 65$ and let $Y_1, Y_2, Y_3$ be the corresponding histogram random variables.

$$\begin{cases} Y_1 = 1_{\{U < 20\}}, & \mathbb{E}[Y_1] = .4 \\ Y_2 = 1_{\{20 \leq U \leq 65\}}, & \mathbb{E}[Y_2] = .5 \\ Y_3 = 1_{\{U > 65\}}, & \mathbb{E}[Y_2] = .1 \end{cases} \qquad (14)$$

Since older people are more likely to prefer to use a phone, the conditional probabilities and corresponding odds are given by

$$\begin{cases} p(X \mid Y_1) = .999, & O(p_{X,Y_1}) = 999 \\ p(X \mid Y_2) = .94, & O(p_{X,Y_2}) = 15.7 \\ p(X \mid Y_3) = .9, & O(p_{X,Y_2}) = 9 \end{cases} \qquad (15)$$

In particular, knowing if they are younger or older is more valuable than the middle range. The Bayes factor (relative odds) expresses the value of knowing the age if someone is willing to bet with the odds $O(p_X)$. So this information allows an expected profit on the bet given by the factor.

$$\begin{cases} BF(X \mid Y_1) = 999/19 = 53 \\ BF(X \mid Y_1) = 19/15.7 = 1.2 \\ BF(X \mid Y_1) = 19/9 = 2.1 \end{cases} \qquad (16)$$

So the value of the information depends on the cases. Finally, if we wish to find the expected value of the information, we take an expectation with respect to the probabilities of the events.

$$\mathbb{E}[BR(X|Y_i)] = 53 \times .4 + 1.2 \times .5 + 2.1 \times .1 = 22 \qquad (17)$$

Some other information about the person may be much less useful in prediction their preference. For example, suppose you know the region where they live and let $Y_1, Y_2, Y_3$ be the histogram random variables. Suppose

$$
\begin{cases}
p(X \mid Y_1) = .03 \\
p(X \mid Y_2) = .05 \\
p(X \mid Y_3) = .07
\end{cases}
\qquad
\begin{cases}
\mathbb{E}\left[Y_1\right] = .3 \\
\mathbb{E}\left[Y_2\right] = .5 \\
\mathbb{E}\left[Y_3\right] = .3
\end{cases}
\tag{18}
$$

Since $\mathbb{E}\left[X\right] = .95$,

$$
\begin{cases}
BF(X \mid Y_1) = 1.9 \\
BF(X \mid Y_1) = 1.1 \qquad \mathbb{E}\left[BF(X|Y_i)\right] = 1.5 \\
BF(X \mid Y_1) = 1.3
\end{cases}
\tag{19}
$$

So with an expected value of 1.5, compared to age, with an expected value of 22, the location information is much less valuable.

