# OpenReview forum: "Empirical confidence estimates for classification by deep neural networks"
_ICLR.cc/2020/Conference — Reject_

### Official Review · AnonReviewer3 · 2019-10-21
**Official Blind Review #3**

**Rating:** 1

**Review:**

Summary: This paper proposes an uncertainty measure called an implied loss. The authors suggest that it is a simple way to quantify the uncertainty of the model. It is suggested that "Low implied loss (uncertainty) means a high probability of correct classification on the test set.". They suggest that the analysis of an implied loss justifies the maximum confidence value of softmax-cross entropy. They also extend to evaluate Top-k uncertainty (the uncertainty whether our prediction is in the Top-5 maximum values of our confidence score or not).

========================================================
Clarity:
I found that this paper content does not seem to be difficult mathematically, however, it is difficult to follow the paper and here I list several parts that can potentially be improved:

1. INTRO: the sentence "Our implied loss interpretation justifies both methods, since we demonstrate that "both these quantities" are uncertainty measures.". What is both quantities here, the maximum softmax probability and something?

2. INTRO: first contribution, accurate estimates of the probability that the classification of the model on a test set is correct. What do you mean by this sentence? I couldn't see the accuracy of the estimation in Table 2.

3. second contribution in INTRO: what is the meaning of "a consistent manner" here. If it means "successfully", is there a case that your method fail? It would be nice to be precise when describing the contribution. Figure 2 is more like a small example but can be considered difficult to convince to the readers about the capability of the proposed method.

4. Figure 1 can be much improved. I found the caption is hard to understand. Loss (y-axis) may be written as Kullback-Leibler loss to be more precise in this context (if I understood correctly).
minor comment: Figure 1(a) [colon missing], since Figure 1(b) has ":" right after. The authors made an effort to explain the color in the caption of Figure 1(b) but the explanation of (yellow) is missing. The explanation of how to train a network to get Figure 1(a) seems to be missing. It seems -log(f_(1)) should be -log(f^{sort}_(1)). Finally, I'm not sure what is P(Correct|x). Is this histogram suggested that when the U_1 is large and the P(Correct|x) is small, we get a correct prediction, or maybe this means the ratio of correct prediction under the value of loss (histogram)? I think it would help the reader to make it more concise. In the description in page 4 -\log(p_{max}) seems never define before, was it a typo?

5.1 First page, last sentence: what do you mean by the current setting. Before this point there is no explanation about the problem setting, only we are interested in quantifying an uncertainty measure of the networks.
5.2 First page, last sentence:  I'm not familiar with Bayes factors, is the last sentence your contribution or it's the finding of the existing work, if it's the latter one, it would be nice to cite them. However, I found this sentence a bit vague: Bayes factor more informative (not sure what is the definition of informative here) better than Brier score under the situation where prob of correct classification is high (is this means high accuracy on the test set?).

6. PRIOR-WORK: Although the authors suggested many existing works, it would be highly useful if the authors discuss the relationship between existing works and their proposed work, e.g., where to put your work in the literature. And since they proposed an uncertainty function, it would be nice to see a few definitions of uncertainty existing works described (doesn't need to be mathematical formulation, I think just an intuitive explanation is sufficient).

7. I am confused with the definition of an implied loss. It is first defined in page 3 with a fixed k (as 1) as a loss where the prediction is correct but in Eq. (4), it looks like a set with one element where y is the maximum prediction score, not a correct label. Then there is U_k(x), if k!=1, is this an implied loss? Although in (5) it is a real number, not a set anymore, I read this paper and took it as a "yes U_k(x) is also an implied loss". Also it would be better to define Kullback-Leibler here to be concise and kind to the readers. Then in def 4.1, it's mentioned that the uncertainty measure U_k(x) is an implied loss if the event has high expected loss, does that mean if the event has low expected loss, it is not an implied loss? My opinion is that the authors may use a definition environment and define precisely what is an implied loss. For example, given "\ell: X x Y \to R, a correct label y \in Y, and an integer k <= K (number of classes), an implied loss is defined as" to avoid confusion.

8. Def 4.1: U(x) without subscript is undefined (perhaps U_k(x)?). What is an element of the set S^\eps_k? If it is a set then what is the meaning of the event S has a high expected loss? Does the definition of implied loss after Eq. (6) and the definition of implied loss in Eq. (5) identical when it is Kullback-Leibler loss?

9. Theorem 4.2: S^\eps seems to be undefined without k. Moreover, how to interpret the bound in (7), it would be nicer to explain the bound after stating the theorem. It is only an inequality that says the left-hand side is smaller or equal to the right-hand side. And does (7) hold for any y? And how tight is the bound?

10. Remark 4.3: what is e_k? what is k here if k in U is set to one? And the name of the remark, how to interpret (8) as "Neural networks are always overconfident?" Is this about neural networks or this apply to any function?

11. Sec 5: Figure 6 is not in the main body but the appendix... If this a mistake (and the paper is supposed to be 9 pages without ref.) or it is supposed to be in the appendix? If It's in the appendix, it would be better to mention that this figure is in the appendix.

12. Sec5: since Bayes factor is highly used in this paper to motivate the use of the measure, I don't think it is a good idea to put the explanation of Bayes factor in the appendix, i.e., it is impossible to understand this paper without knowing Bayes factor.

13: Fig 6: I think it is better and kinder to use U_1, U_5 in the legend of the plot instead of f. What is the model entropy?

14. Table 1: why there is "-" in CIFAR-10, it is better to clarify it in the paper (or maybe I missed it). I am not sure how to interpret the result, if the higher the better, does that mean the Loss is great?, I'm confused with the experimental results.

15. Before the beginning of 6.2: Tables 7 and 6 are in the appendix and we should state clearly it is in the appendix.

========================================================
Comments:
This paper lacks of clarity and difficult to understand. Although it is claimed to be better than existing measure, I am not convinced about that despite many experiments were conducted unfortunately.
For the criticism of using the maximum confidence of the softmax score from the softmax-cross entropy loss may not quantify the uncertainty, it is known theoretically that the score of softmax-cross entropy corresponds to p(y|x) if our prediction function achieve the global minimizer this loss function and our function class to be considered is all measurable functions (Zhang, JMLR2004: Statistical Analysis of Some Multi-Category Large Margin Classification Method). For other losses, see Williamson+ JMLR2016: Composite Multiclass Losses. However, it may not be accurate empirically when we use a deep network as it is reported in Guo+, 2017. Thus, one direction is to do post-processing or finding a way to modify a network. For U_1(k), I feel that it should suffer from the same problem as using maximum confidence score of softmax. Extending to top-k may have a good point when discussing about uncertainty and I believe it is good to explore that direction. For experiments, I would like to know how many trials did the authors run the experiment? and it would be helpful to see the standard deviation of the reported value. I believe this paper can still be improved a lot. For these reasons, I vote to reject this paper this time.

========================================================
Minor comments:
there exists the writing convention of "Top 5", "top 5", "top5". It's better to pick one way to describe it if there is no reason to make it different.
========================================================
After the rebuttal:
I have read the rebuttal.

I appreciate the authors' effort to modify the paper.
Also, please let me state that I totally agree that the problem the authors try to solve is indeed important and relevant for using machine learning in the real-world.

I feel that the structure of the modified version is better than the first version. It is a nice to include the explanation of the Bayes factor in the main body. Appendix C is also very useful for everyone to understand the Bayes factor.

As the author requested, I have read through the whole revised paper (including the appendix) carefully . I am aware of the positive sides of this paper. For example, it is interesting that we can find wrongly labeled data. Utilizing the uncertainty information for several applications. However, I found that the paper still requires a lot of modifications. The authors have modified many of my concerns, but still several of them were not addressed. I also emphasized the comments for parts that are unrelated to clarity (please see below). For these reasons, I decide not to change my score.

Below are my comments after reading the rebuttal (which some of them may be overlapped with the issues that were not addressed in the revised version).
============================
General comments:

1. Although the work is about tackling the empirical confidence estimation problem, Theorem 3.4 and Eq. (5) are providing insights about the population, not finite data points for empirical estimation. If we focus on the population case, it is known that the minimizer of the softmax cross-entropy loss must be a conditional probability $p(y|x)$. Thus, it is natural that as we minimize such a loss, the probability of correct classification must be high, since we can pick the best choice for classification, i.e., argmax of $p(y|x)$. But the most challenging part of the research in this direction is that, although the theory suggests we can get nice confidence information (in population), when we use the deep neural networks, the quality of confidence estimation can be very bad (overconfidence in empirical estimation) compared with the high accuracy we can achieve. As a result, a finer theory that can quantify the quality of confidence estimate for the finite sample case is highly needed, but I think Theorem 3.4 fails to capture this. I am aware that this theorem only concerns the KL loss. I believe that even only for KL-loss, the result can be significant if we discuss a finer theory for empirical estimation.

2. Regarding the Remark 3.5 (Neural networks are always overconfident), in my understanding, the result has no relationship with neural networks at all since it is true regardless of the hypothesis class of interest (e.g., linear models, kernel models, deep networks). This is because it is simply the definition of a loss function (and the result in the paper focus on KL loss, but I believe we can derive for many other losses). We know that the quality of confidence estimation of simple models can be better although the accuracy is worse. Thus, if the objective is to visualize the problem of neural networks, Remark 3.5 does not seem to help and adding neural networks in the remark title can be misleading. Thus, the implication of Remark 3.5 is insufficient to state that Neural networks are always overconfident.

3. What is the advantage of an implied loss? It seems the paper has two separate stories, the first one is implied loss (Sec. 3) and then move on to Bayes factor (Sec.4). Then, there is an adversarial detection problem using the gradient norm in the last experiment. From Sec. 4, the discussion about implied loss is very limited and if I understand correctly, the $-\log p_\max$ and $-log\sum p_{1:5}$ (the latter seems to require a superscript $sort$) are the implied loss, which does not seem to have the clear advantages over other methods. My impression is that the contributions of this paper are unclear and I do not know what is the main point of this paper. While the abstract dedicates most space to highlight the implied loss (nothing about Bayes factor), the conclusion dedicates most space to highlight the Bayes factor. Improving the connection between two parts may help to signify the contributions. Despite all that, I do like the idea of introducing the Bayes factor in this paper.
============================
Clarity:

1. Most importantly, I think the clear and solid definition of implied loss is missing.
According to Definition 3.1, "The uncertainty measure $U_k(x)$ is an implied loss if the event $S^\epsilon_k$ has high expected loss". I believe the implied loss is one of the most important contributions of this paper. I am not sure what does it means by "if the event $S^\epsilon_k$ has high expected loss" How can we define "high expected loss"?. I tried to read the paper many times to understand what exactly is the implied loss, and what is the scope of implied loss (what is and what is not an implied loss).

2. Following my first issue on clarity, what is the definition of uncertainty measure in this paper? According to the paper, it is defined roughly as $U_k$, whose statistics allow us to better estimate $p_k$. And in the abstract, it is emphasized that if uncertainty measure is an implied loss, then low uncertainty means a high probability of correct classification. Is there any uncertainty measure that is not implied loss? Also in the introduction, it seems that both softmax and the model entropy are uncertainty measures, are they implied losses? Is MC-dropout an uncertainty measure or even an implied loss in this context?

3. Eq. (11) left side: I think it is useful to the expectation with respect to which variable, I believe it is $B$. As a result, is it a typo to have $Y_i$ instead of $B_i$ on the left-hand side of (11)?

4. Example 3.3: If we set $k=1$ according to the definition of the implied loss at the last sentence of this example. Will it contradict $U_1(x)$ defined just right before that sentence? Because $y_w$ will become the $2$-nd ranked label, which I feel it is different from $U_1(x)$ defined in Example 3.3

5. I think it is better to clearly state that the figures/tables are in the appendix when referring to them from the main body. I saw the authors refer to Table 5 in Sec. 4.1, Figure 5 in Sec. 4.2 and Tables 6 and 7 in Sec. 4.3. All of them are in the appendix. And it seems some of them are highly needed. For example, the authors said that "by fine graining the bins we can capture relatively small ... on the order of 20" (before Sec. 4.3), then suggested the reader to see Figure 5. I feel Figure 5 must be included in the main body because it is hard to understand that without seeing the figure.

6. I think all figures that have $f_{(1)}$ (Figures 1a, 1b, and 5) must have a superscript $sort$ for all of them. Otherwise, it is wrong.

7. Kullback-Leibler loss is used extensively here without definition. It is important to clarify the clear definition of it. $U_1(x)$ also used extensively for the KL loss case and non-KL loss cases. This can make the paper hard to read. I suggest using $U^{\mathrm{KL}}_1(x)$ when referring to the uncertainty measure with respect to the KL-loss.

8. I saw $-\log(p_\max), -\log(f_{(1)}), -\log(f_1^{sort}), U_1$. Are these all refer to the same thing? And I found that sometimes the argument $(x)$ is ignored in the paper sometimes it doesn't in a quite random way. If they are the same, it would be nice to unify them.

9. The caption of figure 2(b) is uninformative. The authors may consider improving it.

10. It would be very helpful to add the implication or interpretation of the theoretical results to help the readers understand the intuition of the proven results. For example, how does Eq. (3) implies that when small uncertainty implies high chance of correct classification.
============================
Minor typos:
INTRO: "uncertainly measures" -> "uncertainty measures"
5.2: "on-distribution" -> "in-distribution"
Conclusion: logpmax -> write clearly with $$ should be better

Minor comments in the appendix:
1. How does Appendix A related to implied loss or Bayes factor in this paper? Did I miss something?
2. Figure 5:
	2.1 y-axis: is it Bayes ratio or Bayes factor? It seems the Bayes ratio and Bayes factor to be a different thing. Even if it is the same in some literature, I think it's better to use the Bayes factor here for the consistency of this paper.
	2.2 Caption: "entropy" -> "model entropy".
	2.3 Caption: Is it mistakes or do you want to insist on using $U_1$ and $U_5$ in the caption but using different notions in the figure? (in figure, they are $-\log(f_{(1)})$ and $-\log(\sum f_{(1:5)})$).
3. Appendix C: Eq. 16 and Eq. 19: I believe there is a typo. I think it should be $BF(X|Y_1), BF(X|Y_2), BF(X|Y_3)$, respectively.
4. Appendix C: Eq. 18-right: does it need to sum to 1 not 0.3+0.5+0.3 = 1.1?


**Experience Assessment:**

I have published one or two papers in this area.

**Review Assessment: Checking Correctness Of Derivations And Theory:**

I carefully checked the derivations and theory.

**Review Assessment: Checking Correctness Of Experiments:**

I assessed the sensibility of the experiments.

**Review Assessment: Thoroughness In Paper Reading:**

I read the paper thoroughly.

---

> ### Author Response · Authors · 2019-11-11
> **Revised paper should be easier to understand**
>
> Thanks for your close reading of the paper.  It is clear that the paper was hard to read.  The revised paper should be easier to understand.  We hope you will take the time to read it again and let us know if this is better.

---

### Official Review · AnonReviewer1 · 2019-10-25
**Official Blind Review #1**

**Rating:** 6

**Review:**


The paper provides a comparison of several uncertainty measures that have been proposed for neural networks. The goal of the work is to alleviate the lack of clear interpretability of softmax outputs as measures of uncertainty in neural networks for multi-class classification (in particular, here image classification). The authors propose the use of expected Bayes factors as aggregate measures of how much information is conveyed by a measure of uncertainty, and compares several proposed approaches based on this measure on CIFAR-10/100 and ImageNet-1k. The authors also discuss the application of uncertainty measures to detect mislabeled or ambiguous images, detecting out-of-distribution samples and adversarial examples. Passing by, the authors propose a measure based on the norm of the gradient to detect adversarial examples that seems to work well.

There are interesting ideas in the paper. The use of Bayes factors for measuring the quality of the uncertainty estimates, and the comparison of various existing methods on that measure seems useful. The experiments on various tasks also have their own merits.The extension to top-k uncertainty is also interesting.

On the other hand, I found the paper difficult to follow because the contributions are scattered over the paper and the appendices without clear link. A lot of space is devoted to the definition of the implied risk, which does not bring much to the overall interpretation of the results. The criterion for detecting adversarial examples based on the norm of the gradient appears at the end of the paper somewhat independently from what was before, and the definition and computation of the Expected Bayes factor is entirely deferred to Appendices, which makes the main paper not really self-contained.

The paper also lacks a conclusion. The various uncertainty measures seem to perform differently depending on the datasets, and it is unclear what the authors recommend in the end.

Overall, it seems to me that with a bit of restructuration, the paper would be an interesting contribution -- for instance in terms of the assessment of dropout variance vs direct model entropy. It seems to me though that for now the paper lacks coherence and clarity in the message.


====== after author rebuttal

The rebuttal answer most of my concerns, and I raised my score to weak accept. Overall, even if there is no clear evidence that one method to evaluate uncertainty is consistently better than the others, I feel that the idea of using Bayes factors to compare uncertainty measures is possibly impactful.


**Experience Assessment:**

I have read many papers in this area.

**Review Assessment: Checking Correctness Of Derivations And Theory:**

I assessed the sensibility of the derivations and theory.

**Review Assessment: Checking Correctness Of Experiments:**

I assessed the sensibility of the experiments.

**Review Assessment: Thoroughness In Paper Reading:**

I read the paper at least twice and used my best judgement in assessing the paper.

---

> ### Author Response · Authors · 2019-11-11
> **We took your advice: more streamlined and added conclusion**
>
> Thanks for suggesting these improvements.  We did what was asked and the paper has turned out much better.   As explained in the general comment
> 	•	streamlined sections 3 and 4 to be shorter and more clear.
> 	•	we moved adversarial examples to the appendix, since this is less connected to main point
> 	•	moved the definition of Bayes factor to main body, and included a detailed example calculating it for Model Entropy on ImageNet
> 	•	we added a conclusion, excerpted above.
>
> We hope that you will find that the paper overall is improved.

---

### Official Review · AnonReviewer4 · 2019-10-31
**Official Blind Review #4**

**Rating:** 6

**Review:**

This paper shows that the maximum softmax probability is useful for uncertainty estimation on in-class data and not just for detecting out-of-distribution data. They argue this with the Bayes Ratio, which here is an uncertainty estimation quality measure that seems worth exploring more for assessing the quality of uncertainty estimation techniques. The community is still in need of a good uncertainty estimation measure (Brier score is too tangled with accuracy, has low numerical resolution, and hardly penalizes consistent overconfidence; AUROC for error detection doesn't budge; the Soft-F1 score is without theoretical motivation; area under the response rate recall curve is too close to 1 when accuracy is high), and this could be it. They evaluate on ImageNet-1K, which most uncertainty estimation papers fail to consider. By considering CIFAR-10, CIFAR-100, and ImageNet, we get fuller picture of the ranking of the utility of many uncertainty estimators, which is somewhat important for the community. This paper is currently borderline, in that what it proposes is simple, but perhaps too simple. Its empirical contributions are fairly minimal, though its proposal of the Bayes Ratio for uncertainty estimation quality assessment could quite impactful.

Smaller Notes:
Figure 1 should be on page 3 not 2.

Some content in Appendix B should be incorporated in Section 5.1

> In the odds have increased
If

B.3 should have a worked example in our setting as well
Eqn (19) should say > 65 not <65

Notation of Y is confusing when it means a binning of U. Perhaps use "B"?

Is the histogram adaptively computed? Is that what “chosen to have equal weight” means?

Perhaps compare the Bayes ratio with RMS Calibration Error, AURRA, or AUROC for error detection.

Update: The other reviewers are concerned about lack of clarity which is separate from why I like the paper.

**Experience Assessment:**

I have published in this field for several years.

**Review Assessment: Checking Correctness Of Derivations And Theory:**

I carefully checked the derivations and theory.

**Review Assessment: Checking Correctness Of Experiments:**

I carefully checked the experiments.

**Review Assessment: Thoroughness In Paper Reading:**

I read the paper at least twice and used my best judgement in assessing the paper.

---

> ### Author Response · Authors · 2019-11-11
> **Additional clarifications**
>
> Thanks for the good advice.  In addition to the revision discussed above, we also responded to the smaller notes.
>
> Added the sentence:
> In practice, we defined the bins to have an equal number of examples in each bin.  In addition, in order to make the histogram Bayes factors finite, we require that each bin have at least one correct and one incorrect example.
>
> Added to the introduction:
> Compared to other methods (such as AUROC or Brier scores) Bayes factors better distinguish improvements to confidence for methods which are already quite accurate, as is the case for top 1 or top 5 uncertainty for image classification.
>
> Added:
> The Bayes factor is a way to measure the value of new information, in terms of how much the expected winnings of a fair bet increase, when the information is available.   Unlike other  measures of confidence, which are additive, the Bayes factor is \emph{multiplicative}.
> On a model which is correct 95\% of the time, there still a lot of value in knowing when the probability correct increases to 99.5\%.  In this case, the Bayes factor is close to 10.   On the other hand, going from 50\% to 54.5\% gives a Bayes factor close to 1.2.  On the other hand,  additive scoring methods give equal weight to both improvements.   As an example, we show in Table~X that the Brier scores of eight different measures of confidence all lie close together, between .033 and .076.  On the other hand, the expected Bayes factors range more widely, from 1.3 to 16.6.

---

### Author Response · Authors · 2019-11-11
**Revision for clarity in repsonse to referees**

Based on suggestions of referees, we restructured the article to make it more readable.
	•	Moved the definition of Bayes factors into section 5.
	•	Added an example of calculation of Bayes Factors using the Model Entropy on ImageNet. (Example 1, quoted below)
	•	Added a conclusion section (quoted below)
	•	Moved adversarial attack detection to appendix for space.
	•	Tightened up Sections 3 and 4, by merging into one section, Uncertainty measures and estimates, making it shorter and easier to read.

The entire paper is slightly shorter, and more clear.

Example 1:
This example follows closely the confidence bins for top 5 on ImageNet-1K, using the Model Entropy, as in the first row of Table~X.
Consider a model with $p_X = .94$.  In this the odds are $94$ to 6, so $O(p_X) = 15.6$.
Define three bins for Model Entropy with bin edges $0.31, .140$.  Then with probability .55, data is in the first bin, in which case the probability correct is .99.  So the Bayes factor in this bin is $(.99/.94)(.06/.01) = 6.3$.  Thus knowing the Model Entropy is less than $.31$ tell you that you are 6.3 times more likely to be correct than on average.   The second bin consists of data with with Model Entropy between $.31$ and $.14$, which occurs with probability .31, in this case, the Bayes Factor $(95/94)(6/5)= 1.2$ is nearly one, so there is little additional value to knowing data is in this bin.  Finally, when the Model Entropy is greater than .14, which occurs with probability .14, the probability correct is only .8.  In this case, the relative probability to correct is worse, to the Bayes Factor is given by $(94/80)/(6/20) = 3.9$.  The expected Bayes factor is the weighted average of the Bayes factor of each bin, weighted by the probability of the bins
\[
 \mathbb E {{BF}(X \mid B_i )} = 6.3\times .55 + 1.2\times .31 + 3.9\times .14 = 4.4
\]
So the expected value of the Model Entropy, for the chosen bins, is 4.4.   By fine graining the bins we can capture relatively small  and relatively large values of the Model Entropy which can have Bayes Ratios on the order of 20, see Figure~X.  Thus the expected Bayes ratio with 100 bins is 8.18, as shown in Table~X.

Conclusion:
With the goal of using measures such as model entropy as a surrogate for the (unknown) model loss, we defined confidence measures as random variables which are large when the loss is large.   Using this definition we proved that confidence variables can be used to estimate the probability that a model low expected loss makes a correct prediction.   In practical terms, this amounts to defining a confidence measure (such as model entropy or log pmax) and binning the values.

We presented the expected Bayes ratio as an effective measure of confidence.  Since models are already very accurate, it is important to measure the relative confidence.  The Bayes factor is a multiplicative factor to the probability (or odds) that a model is correct.  For example, showing the that model entropy on ImageNet with 100 bins is 8 means that knowledge of the model entropy (and the bayes factors for the bins) allows us to predict the probability that the model is correct 8 times more effectively.

The Bayes factors was used to compare existing confidence measures on different tasks.  The main task was estimating the probability of a correct prediction on the images from the data set.  Additional tasks included: detection of off manifold data, detection of adversarial examples, and  detection of mislabelled images.  The latter were found by searching for highly confident predictions which were labelled incorrect.

---

### Decision · Program_Chairs · 2019-12-19

**Decision:**

Reject

**Comment:**

The paper proposes to model uncertainty using expected Bayes factors, and empirically show that the proposed measure correlates well with the probability that the classification is correct.

All the reviewers agreed that the idea of using Bayes factors for uncertainty estimation is an interesting approach. However, the reviewers also found the presentation a bit hard to follow. While the rebuttal addressed some of these concerns, there were still some remaining concerns (see R3's comments).

I think this is a really promising direction of research and I appreciate the authors' efforts to revise the draft during the rebuttal (which led to some reviewers increasing the score). This is a borderline paper right now but I feel that the paper has the potential to turn into a great paper with another round of revision. I encourage the authors to revise the draft and resubmit to a different venue.